# The Gender of Retirement in a Double-Ageing Country: Perspectives and Experiences of Retired Women and Men in Portugal

**DOI:** 10.3390/bs13090774

**Published:** 2023-09-16

**Authors:** Maria Carolina Pereira, Miriam Rosa, Maria Helena Santos

**Affiliations:** 1Iscte—Instituto Universitário de Lisboa, 1649 Lisboa, Portugal; miriam.rosa@iscte-iul.pt (M.R.); helena.santos@iscte-iul.pt (M.H.S.); 2Iscte—Instituto Universitário de Lisboa, CIS-Iscte, 1649 Lisboa, Portugal

**Keywords:** ageing, retirement, gender

## Abstract

This article aimed to explore the perspectives and experiences of women and men at the stage in their lives following professional retirement, enriching the present and future of a continuously ageing Portuguese society. In order to better capture the diversity and complexity of each individual’s experiences, a qualitative methodology was used. Semi-structured individual interviews were carried out with sixteen retired people, eight women and eight men, aged between 59 and 88 years old. A thematic analysis allowed us to identify five themes in the interviewees’ discourses. We concluded that gender may be a source of heterogeneity at this stage of life, suggesting that these findings should be analysed in the scope of a life course perspective, and highlighting the impact that the different trajectories of women and men have, as well as the historical and social context in which they take place.

## 1. Introduction

From the second half of the twentieth century, demographic changes have been recorded at the level of population ageing resulting from the decline in mortality and fertility rates, and the increase in average life expectancy [1]. Currently, Europe has the highest percentage of population aged over 60 years, and Portugal is not excluded from this demographic scenario dominated by ageing. Recently, the provisional results of the 2021 National Census point to “double ageing”, characterised by an increase in the elderly population and a decrease in the number of young people. According to the National Statistics Institute (INE), 23.4% of the population is over 65, as opposed to younger people (up to 14 years old), who represent just 12.9% of the population [2]. In addition, the ageing index shows that there are 182 elderly people for every 100 young people, compared to 125 in 2011. These figures mean that Portugal is currently among the five oldest countries in Europe [3].

Thus, the study of the ageing process and its representative events, such as job retirement, has been receiving scientific attention [4], aiming at understanding when and why people retire, what they do, and what they actually experience after retiring [5].

Research on this topic aims to provide important findings for the retirement literature in general and from a gender perspective in particular. Despite the increasing number of studies in this area, little is known about how retirement perspectives and experiences are related, and what the similarities and differences are between women and men at this stage in their lives. In fact, Kim and Moen in 2002 [6] alerted researchers to the fact that most of the existing studies in this area focused only on men’s retirement, and there was little information on women’s experiences. Surprisingly, around 20 years later our knowledge about the role of gender in this area is still quite limited [7]). This knowledge is relevant for several reasons. It is known that there are gender differences in earlier stages of life, regarding, for example, the uses of time [8] and the working sphere itself [9,10]. Obtaining data of this nature is even more relevant in Portugal, where, in general, there is a scarcity of studies on retirement, despite the constant population ageing [4]. In addition, this research aims to make important contributions to the work of professionals and companies. In a labour-oriented society, retirement can bring added challenges. In this sense, studying retirement will allow for a better understanding of this reality, making it possible to create individualised measures and interventions for both men and women that favour their well-being and a good transition and adaptation [11]. Finally, studying this topic will also help to demystify the mistaken beliefs that exist in relation to this stage of life, which is still neglected and considered one of the least interesting stages of human life [4], since the lack of knowledge about the process of growing old leads to negative attitudes towards it [12]. A growing body of authors has demonstrated how older people are a target of ageism, i.e., negative attitudes and practices based on the age of individuals, e.g., [13,14].

In summary, this paper seeks to bridge these gaps by empirically exploring men’s and women’s perspectives and experiences of retirement from a gender perspective.

### 1.1. Retirement and Its Background

In this research, we will analyse retirement as the complete and permanent exit from paid employment [15].

Currently, retirement is perceived as a complex and multidimensional process, which occurs over a period of time, and not only as a single event [16,17]. Retirement can be characterised through several dimensions, namely, in relation to its voluntariness (voluntary vs. involuntary), the number of hours per week (partial vs. total), and the time of retirement, which is related to the retirement age (early vs. on time) [18]. For example, in Portugal, according to the national social welfare, the legal age for retirement has increased and in 2023, it corresponds to 66 years and 4 months [19].

Several studies have analysed the antecedents of retirement. Findings have highlighted a multitude of individual, family, organizational, and social factors that interact and influence the decision to retire [17,20,21,22]. For example, at the individual level, health and financial situation are the most significant predictors of the decision to retire. People with a declining health status may wish to retire early because they perceive that their working capacity is impaired, and retirement is a great opportunity to recover [22]. On the other hand, many workers may eventually continue to work beyond their normal retirement age because of the possibility of getting a higher pension [23]. In relation to the characteristics of the work context, individuals who have high levels of job satisfaction may ultimately postpone the decision to retire [20].

### 1.2. The Transition to Retirement

Throughout the normal course of human development, there are numerous transitions, retirement being one of them. Living in a society characterised as work-oriented, the retirement period may equate with identity crises [4]. This transition is based on two crucial challenges: adapting to the loss of the work role/status and the social interactions that characterised the person’s life as an active worker, and developing a new lifestyle [24], characterised by an increase in free time and, sometimes, a decrease in income [25].

### 1.3. Life Course Perspective

The life course perspective has been used over the years to guide several research studies focusing on ageing. This perspective is distinguished by its focus on time, context, and process [26], allowing researchers to analyse how trajectories influence ageing, as well as the historical and geographical contexts in which it takes place [27].

In the present research, we demonstrate how a life course perspective can promote a better understanding of the study of gender and retirement, focusing, in particular, on different trajectories that women and men have in this life stage, as well as the context in which they develop. According to this perspective, gender constitutes one of the variables influencing the trajectories along the life course [28] and, consequently, a source of heterogeneity of the retirement experience [6].

### 1.4. Gender: The Different Trajectories of Men and Women

Gender studies’ discussions have warned of a differentiation between gender (a socially constructed dimension) and sex (a biological dimension) [29,30]. In this article, we address gender as masculine and feminine, circumscribing it to social attributes and opportunities associated with being a woman and being a man, as well as the social relations that may arise between both (e.g., relations between women, relations between men, and relations between men and women) [29]. Women and men thus present different social roles and different trajectories and life paths, which should not be ignored. The literature on the uses of time has made extensive contributions, highlighting the existing inequalities between men and women, but focusing mostly on working age [8].

In the so-called Western society, paid activities have become the greatest source of occupation and recognition, due to the possibility of generating monetary value. This idea leads to multiple biases, determining that other human activities, no less important, are continuously devalued, and that those who undertake fewer hours of paid work are considered to have less work performance and more “free” time [8,31]. In particular, women are largely discriminated against in this process, as they perform a multitude of unpaid tasks, such as domestic and caregiving tasks, which compromise their availability for paid work. Even if some men perform some type of unpaid work, male participation is lower than female participation [32], which has not improved with the pandemic [33,34], bringing consequences for the women’s life course [8,31], including in the post-paid-work life phase [35].

In short, in post-industrial societies, men are more likely to be hired for jobs, especially for positions of authority and leadership [36]. Women, in contrast, have quite distinct career paths and are not only much more likely to take breaks, but also to work fewer hours, and engage in unpaid work due to their domestic and caregiving responsibilities, not to mention their under-representation in senior positions [36,37]. Although there are few studies that address these issues in the Portuguese context, data from the national time-use survey in 2015 reported that in all age groups, women devoted more time to household chores and care [8]. In addition, according to the 2021 statistical bulletin from the Commission for Citizenship and Gender Equality (CIG), women in European Union countries, including Portugal, continue to make up the overwhelming majority of the inactive population due to caring responsibilities [31].

Until the 1980s, many researchers considered that retirement was not significant for women, as paid work only had a central role in men’s lives [38]. Calasanti (1996) first highlighted that theoretical knowledge about retirement was based on men’s experiences and therefore could not be generalised to women’s experiences [39]. h, retirement experiences need to be understood through the different experiences of men and women [39]. For just as men and women have different work and life trajectories, in general, they also have different ways of adapting to and experiencing retirement [6,24].

Research has shown that gender differences existing during active working life are perpetuated in retirement [40], and that women’s decision to retire is more informed by financial needs [41]. While men tend to describe retirement as freedom from responsibilities, women mention the freedom to plan domestic work [42]. As a result of lower labour market participation throughout life, women, compared to men, face various difficulties when it comes to retirement, and they often work longer and find themselves in a more precarious financial situation than men [37].

Although a substantive body of knowledge on gender and retirement is emerging, little or nothing is known about the perspectives and experiences of men and women in retirement. Recent research on retirement and gender, concerned with the sustainability of social policies and pension systems, has concentrated its focus on health outcomes such as mortality rates or disease/well-being associated with retirement. Another area of great interest concerns economic/financial aspects such as retirement planning, or working after retirement. Thus, there is a gap regarding psychological accounts on retirement experiences [43,44,45]. The relevance of the present research increases in the Portuguese context, since (i) Portugal is one of the countries with the highest age average in Europe [3]; (ii) there is a lack of studies on retirement in this context [4] (for a recent exception, see [46]); and (iii) Portugal continues to have deeply rooted traditional gender roles [9]. Also, despite the existence of studies addressing these gender inequalities in working life in Portugal, e.g., [8,47], there is no knowledge on how these inequalities may be perpetuated in and after retirement. Thus, from a gender perspective, this research aims to understand how people experience their retirement, empirically exploring men’s and women’s perspectives and experiences regarding their retirement. More specifically, it aims to analyse (i) the reasons that led participants to retire; (ii) their perspectives on retirement; and (iii) their current experiences.

## 2. Materials and Methods

### 2.1. Participants

Individual semi-structured interviews were conducted with sixteen retired people, eight women and eight men, aged between 59 and 88 years (M = 73; SD = 8). All interviewees are Portuguese and live in the district of Leiria, Portugal; most of them belong to rural areas. Their retirement age is between 55 and 68 years old (M = 61; SD = 4), and their retirement time varies a lot more, between 2 and 22 years (M = 12; SD = 7). Table 1 shows the sample characterization regarding their type of retirement, retirement time, reason for retirement, and how their time is used.

### 2.2. Procedure

The study met all the ethical principles of research and personal data processing (e.g., informed consent and debriefing, including information on the voluntary nature of participation, processing and storing personal data, confidential treatment of information, etc.), having obtained a favourable opinion from the Ethics Committee of the University (48/2022). The process of recruiting participants was initially based on convenience, as people we knew in retirement were contacted by telephone and in person. We then used the “snowball” technique, with participants being asked at the end of each interview to make contact with friends and/or acquaintances who might be interested in taking part in the study. The only inclusion criteria were speaking Portuguese and having been retired for at least two years. All the interviews were carried out face-to-face by the first author of the present paper, in quiet locations, depending on the preference of the participants. The data collection process took place between 16 February and 22 March 2022, with each interview taking an average of 22 min. The final number of interviews was determined by the saturation criterion, i.e., when the new interviews began to show a number of repetitions in their content [48].

### 2.3. Instruments

Data collection was carried out using two distinct instruments: a brief sociodemographic data questionnaire and a semi-structured individual interview script. This includes eight questions duly supported by literature review and divided into three main thematic dimensions: the career path and professional context; the retirement antecedents; and the perspectives and experiences of retirement.

### 2.4. Data Analysis

Thematic analysis [49] was the methodology used to analyse the corpus of textual material from the 16 interviews. This is a qualitative methodology widely used in psychology and other fields, allowing patterns (themes) within the data to be identified, analysed, and reported. The overall themes and their sub-themes capture something important about the data in relation to the research question, representing a certain level of standardised response or meaning from the data set. In this sense, thematic analysis is understood as a “flexible and useful research tool which can potentially provide a rich and detailed, yet complex, account of data” [49] (p. 78).

Thematic analysis encompasses six distinct phases, which form part of a recursive process that should not be rushed at any point. Thus, over time, as researchers, we must allow ourselves to move either between the different stages or through the data set. The six phases are as follows: (i) familiarization with the data; (ii) generating initial codes; (iii) searching for themes; (iv) reviewing themes; (v) defining and naming themes; and (vi) producing the report; see [49] (p. 87).

In the present research, we used a mixed thematic analysis, namely, inductive and deductive. We followed a largely deductive logic, driven by prior engagement with the theoretical framework on the topic [49] of retirement from a gender perspective. However, we also followed a partially inductive analysis by opening up to the possibility of identifying unexpected themes, considering the theoretical framework. This recursive analytical strategy was carried out among the research team, ensuring that themes provided a rich description of the whole body of data and reducing the likelihood of overlooking insightful nuances.

The results are presented in the following section.

## 3. Results

The thematic analysis allowed us to identify a set of themes and sub-themes in the discourses of the 16 people interviewed, as shown in Table 2.

### 3.1. From the Absence of Gender Perspectives to Men’s Perspectives on Retirement

In this theme, three sub-themes were identified: (i) lack of perspectives on retirement; (ii) shared perspectives—freedom and travel; and (iii) men’s perspectives on retirement.

A first sub-theme encompasses the discourses of interviewees who did not reveal prior perspectives on retirement, stating, for example, that they never thought of retiring so early, that they do not usually think much about the long term, or that they simply did not think much about it, as the following excerpts illustrate:

I never imagined it, because for one thing, I had never imagined that I would retire so early. (I9, woman, age 59)

I guess I never really thought about it, I never think about things that way in the very long term. (I10, man, age 66)

A second sub-theme focuses on the consensus between the women and men interviewed in terms of perspectives that were quite centred on the issue of freedom and willingness to travel:

I imagined it so many times, I always said (…) that I’m going to retire at 50, from then on I’m going to travel, I’m going to know the world. (I12, woman, 68 years old)

I imagined my life in retirement with more trips, getting to know more of Portugal, being able to go here and there, having more availability. (I6, man, 87 years old)

A last sub-theme highlights men’s perspectives on retirement, showing that they planned to dedicate themselves to tasks more related to being outdoors, such as, for example, agriculture, gardening, and professional activities:

I then thought the following, I had this piece of land here, I was going to plant an orchard here, I have water there and I started planting some plants there, some peach trees, pear trees, lettuces and strawberries. I thought I was going to lose a few hours a day here and that’s what happens. (I3, man, 72 years old)

### 3.2. From Gendered Occupations in Retirement to Shared Occupations

A second identified theme shows that most of the occupations of the people interviewed are gendered, although there is one occupation that is shared, giving rise to three types: (i) feminine occupations; (ii) masculine occupations; and (iii) a gender-shared occupation (gardening), detailed here in three subthemes.

A first sub-theme allows us to conclude that, at retirement, women continue to take on more housework and family-care-related activities. In general, women have a more restricted day-to-day life than men, being more circumscribed to the domestic sphere than men, and devoting little time to socializing and hobbies:

I have more time with my grandchildren, I can see them grow up and help my daughter too (…). I pick them up from school, I support my daughter at home, and I socialise with some friends I have, there are not many at the moment, but that’s what I do, I also read some books. (I2, woman, 71 years old)

As I already told you, it’s waking up doing my housework, a lot of times, I go walking first, I have water aerobics twice a week and every other week I go to Lisbon (…). I expected to travel, not to be here at home, taking care of the house. (I12, woman, 68 years old)

A second sub-theme shows that the men interviewed show not only a greater diversity of time uses, but also a greater amount of time spent on socializing activities, hobbies, or doing some occasional work within their prior job, or another job. In general, most of their time is spent outside:

I started retirement in 2001, exactly as an account manager at Santander [bank], and since then, besides being retired, having some leisure, walks and things like that, I have a small agriculture, where I keep myself busy(…). I also continue to work with the same bank in the commercial area (…). Once I retired, I started to get up a little later, between 8 and 9 o’clock in the morning, I walk around the block, drink a coffee with friends, talk a little and then I go back home, shave, get ready to go to lunch either with a son, or with a grandson, or with both. (…) And then I have an occupation that I like very much, that I always liked, and that professional life did not allow me, which is music. (I1, man, 77 years old)

The third sub-theme highlights the existence of an occupation shared by both men and women—gardening:

And I also go shopping when I need to, I garden a little bit today and a little bit in a week’s time. (I9, woman, 59 years old)

Then, besides farming, I also have gardening, I’m the one who does all this here. (I3, man, 72 years old).

### 3.3. Gender Differences in Perceptions of Retirement Pensions

The third identified theme focuses on perceptions about retirement state pensions (monetary value), again noticing gender differences regarding this issue. We distinguish two sub-themes, perceiving that (i) women consider their pensions as being low, and (ii) men are more satisfied with their pensions.

A first identified sub-theme reveals that, in general, women do not consider the value of their retirement as being adequate, not only considering the time they have worked, but also because of the cost of living and the needs they face in retirement:

There is a counter to retirement, the pensions are very low even though I already had 39 years of discounts [for the state social welfare], I was a top painter for so many years, with a top salary and I was even in a leadership position (…). If I didn’t have a husband and my own house, maybe, with what I spend on medication, it wouldn’t be enough. (I9, woman, 59 years old)

Another identified sub-theme shows that, although there were also some men who admitted they would like their pensions to be higher, we could perceive that, in general, they were more satisfied with their pensions than women were:

So my expectation was, in a way, a little bit above average, given that the sector I was in was a profitable sector, it was a safe sector, and so I always aspired that if I didn’t have any bad luck in life, that when my time came, it was going to have a reasonable retirement (…). I have a retirement pension slightly above average, which really allows me a relaxed life. (I1, man, 77 years old).

### 3.4. Gendered Experiences of Retirement

The fourth identified theme shows gendered experiences of retirement, with men experiencing retirement in a more positive way than women, who reported several negative experiences at this stage. Thus, we highlight two sub-themes, (i) women’s experiences; and (ii) men’s experiences.

The first sub-theme concerns the experiences of women, who admitted that retirement was not meeting their expectations. They reported a predominance of negative experiences at this stage, connected with ageism [i.e., age-based negative attitudes; e.g., [14] and the respective negative representation of age that predominates in our society. However, we can also perceive some positive experiences reported by these women, who highlighted the time spent with their grandchildren:

Retirement always scared me because it was significant of old age. I, as a retired person, feel sad, because we feel more that age has passed, many [symbolic] movies have passed in our lives and those movies will be much shorter from now on (…). (I4, woman, 72 years old)

The ideal would be for me to feel well and not have this problem, but, well, if I didn’t have grandchildren, I think I’d go into a tailspin [downward spiral], so, as they take a lot of my time, I’m always happy and well, they help a lot with the psychological part. (I9, woman, 59 years old)

Another identified sub-theme focuses on the experiences of men, and it is important to note that all men interviewed stated that life after retirement is meeting their expectations, immediately highlighting the existence of positive experiences. Another aspect frequently mentioned by men was the freedom that retirement gave them, which is one of the main sources of positive experiences. This freedom also allows them to create a new routine, different from the one they had during most of their adult life. In summary, we can say that, in general, retired men feel free, relieved, uncommitted, joyful, and satisfied with life:

That, of not being subject to anyone’s orders, you have no idea, for me, it’s a thing which comes from another world, there’s nothing that can pay for that freedom. You don’t have to be accountable to anybody. When you are working, you are accountable, you obey certain rules and you have to follow them, or not, but if you don’t follow them, you suffer the consequences. But you are having the possibility and the freedom to get rid of that, it’s something from another world. (I10, man, 66 years old)

### 3.5. Obstacles to an Active Retirement

The fifth, and final, identified theme focuses on obstacles to active retirement, highlighting the existence of two sub-themes: (i) decline in physical and psychological health; and (ii) negative impact of the COVID-19 pandemic.

A first sub-theme encompasses the issues related to health status. There is a clear consensus among the respondents, with several reporting a decline in their physical and/or psychological health in recent years. These declines are addressed as an impediment to active retirement, namely, the accomplishment of tasks, plans, and prospects they held:

You know, when I used to drink coffee, now I can’t drink it, what time would I go? At 6 o’clock in the afternoon, when nobody was there, I used to smoke my cigarette, now I can’t smoke. Look, I stopped smoking there in the “cabeço” [small hill], I went to dinner with colleagues, there were five of us, I smoked four or five cigarettes in a row, that day, we were chatting, and I was smoking, I got home it was almost 1 am late night and suddenly a heart attack, I had to go to the hospital. (I12, woman, 68 years old)

I had this and the [inner ear] crystals not going into place, once or twice on the way out of the rehearsals they had to get me back in my house, so then the doctor told me right away, okay, suspend that indefinitely and okay. It cost me a lot, because I really liked to belong to the academy choir, we sometimes did those shows, here in Benedita. (I16, man, 77 years old)

A second sub-theme depicts the negative impact of the COVID-19 pandemic on the lives of the people interviewed. In fact, in this regard, there is a consensus, with most participants reporting a decrease in their routine outside with the pandemic, compromising an active retirement. This decrease, essentially, occurred through a reduction in their routines, and in the time allocated to socialization activities and hobbies. In essence, the pandemic increased the loss of a sense of freedom, leading to fear, fatigue, and social isolation:

The first two years, I was quite active, I went to the senior university, I went on walks, hikes, I went on a lot of walks with the university, but then we went into pandemic and since then it was a meltdown. I’ve never done anything that I like, I’ve never had any activities that help me and, since then, I’ve been dedicated to my garden (…). (I14, woman, 68 years old)

Since the pandemic, I also started to avoid going downstairs. I get up, drink a cup of coffee, do my little things, if I have time, I go to the backyard and entertain myself. Now I spend more time at home because we can’t be too careful, with this pandemic, that’s how it is, we have to face this as reality and not expect miracles. (I16, man, 77 years old)

## 4. Discussion

The results of this research offer important developments for the study of gender and retirement, increasing our knowledge about how men and women live during retirement in Portugal. It was possible to see that the interviewees retired due to a variety of reasons, including reaching length of service, the opportunity arising, general fatigue, fatigue from a factory environment, exhausting work, health problems, and the end of employment subsidy time. Although most respondents reported only one reason behind their choice to retire, we recognise, in light of findings from other studies, e.g., [17,20,21,41], that the timing of retirement is not influenced by a single factor alone, but rather a multitude of aspects that interact and influence this decision. Although there may be more reasons underlying the decision to retire, these results allowed us to identify a multiplicity of individual and organizational reasons why people retire, allowing us to meet our first research objective and showing that retirement is a complex process influenced by several factors [21].

Regarding the perspectives on retirement, some of the respondents reported having no perspectives on retirement, in line with some studies, for example, [50], where they state that uncertainty about this phase of life is a real situation for workers and should not be ignored, since retirement planning is closely linked to positive experiences of satisfaction [51] and well-being [52] in retirement. Still, there are other reasons that may help to understand this scenario, namely, the existence of unanticipated and involuntary types of retirement reported by some interviewees, such as disability retirement and early retirement due to long-term unemployment. Additionally, we believe that the absence of expectations can also be understood through the lack of adequate planning by employers. Today, we know that a retirement plan brings benefits not only to employees, but also to employers. For example, by investing in retirement planning, they help to attract and retain employees, demonstrating a commitment to their long-term well-being and reducing the costs of new hires and training. Finally, it is relevant to mention that those people who are in worse economic and health conditions may also have fewer resources, limiting their choices and retirement planning [53].

On the other hand, several respondents presented perspectives focused on the issue of freedom and availability to travel. These results are not new, since the end of professional life is often desired, as it means freedom of time use and the fulfilment of personal interests [4]. Additionally, according to Atchley (1989) [54], most people refer to leisure as one of the major motivations for the retirement transition (for a more recent account, see [41]). Finally, we also obtained results that allowed us to identify the existence of men’s perspectives regarding this issue, namely, that they planned to engage in agriculture, gardening, and professional activities. The existence of these perspectives of outdoor activities by men, as opposed to other types of tasks (e.g., housework or caregiving), can be justified through the gender expectations that are present throughout the life course and that also perpetuate in the way individuals envision retirement [39].

Regarding the second objective of our research, these findings allow us to see that there is still a great lack of planning regarding this part of life, both by men and by women. In addition, leisure and freedom emerge in both groups as one of the perspectives they have for retirement, although men also present different perspectives, linked to outdoor activities. In sum, we can see that there is great variability in the way men and women plan their retirement, in line with other studies [53,55].

What is particularly insightful about our results is the interplay between people’s perspectives and their actual experiences during retirement. While recent research from a gender perspective tends to focus on retirement planning or in retirement outcomes [50,56], the participants in our study engaged in meaningful processing about their current standing, with women falling shorter in terms of their expected accomplishments.

Regarding occupations during retirement, the interviewees’ responses highlighted and accentuated the existing differences between men and women. The literature on uses of time [8,31] allows us to understand that gender differences existing in the active life phase perpetuate throughout life trajectories, including this later stage of human life. There is not only the idea that men and women use their time differently, but also that there is a discrepancy in the division of labour, which extends to various stages of life [57]. In general, most occupations in retirement are gendered, with women being more occupied with household chores, family care work, and physical exercise, as opposed to men, who occupy more of their time with hobbies and socializing activities. Decades later, the results are still in line with Calasanti’s (1996) [39] study, which showed that, also during retirement, women are expected to continue engaging in housework and care work, assuming a disadvantageous position in relation to men. Indeed, while hobbies and other socializing activities are optional activities, housework and caregiving are duties and responsibilities that women must often undertake.

Additionally, men not only have a greater diversity of occupations, but also spend more time outside the house, as opposed to women, who have a more restricted day-to-day life. The fact that the concerns characterizing women’s active working lives remain unchanged in retirement [58], may help to understand why they spend less time on socializing activities and hobbies. In sum, these findings help to highlight the expectations and gender roles at this stage of life, where women continue to be busy with domestic, family, and caregiving tasks, while men have the freedom to build a new routine different from the one performed for most of their lives. These results are in line with the results of the national survey on uses of time, which reported the existence of inequalities between men and women in the division of household chores and care work in Portugal, also in the age group over 65 years old [8].

If, on the one hand, time use is influenced by gender, it could also be influenced by other factors, namely, the Portuguese historical context and the geographical dimension where respondents live [35]. In Portugal, traditional gender roles, are still very much ingrained. In fact, according to Amâncio and Santos (2021) [9], the social changes that took place after the transition to democracy were not accompanied by a gender equality ideology. On the other hand, some occupations reported here are carried out because interviewees live in a rural context where outdoor activities, such as farming and gardening, are possible. This would probably not characterise the daily lives of all participants if they lived in an urban context.

Regarding how the respondents perceive their retirement pensions, men were much more satisfied than women, who consider the value of their pensions to be low. These results are in line with Kim and Moen’s (2002) [6] study, which showed that women considered their pensions more inadequate than men and objectively lower as a general trend [45]. Again, different life and work trajectories may help to understand these findings. The fact that women are more likely to take career breaks, engage in unpaid work (e.g., housework and caregiving), and be under-represented in high-ranked positions [36,37], may cause them to have lower pensions. According to Duberley et al. (2014) [37], women generally have a harder time than men when the time comes to retire, since they are in more economically disadvantaged positions than men. This issue also leads to a discussion over the fact that unpaid work, such as domestic and care work, is not considered or valued in our society [59], leading to retirement pensions being based on a notion of work that does not consider the domestic and care activities that women perform throughout their lives [60]. Supported by the recent literature, we would have expected to find accounts of coordination between dual-earner couples, i.e., interviewees mentioning a coordination among the couple on who invests in breadwinning (typically the man) and who invests in caregiving (typically the woman), both before and after retirement [59,61]. However, we did not find that either in the narrative about the pensions or the retirement planning. We can speculate that those decisions were not discussed as a couple, or that when elaborating on their experiences, participants did not consider to a great extent the weight of being embedded in a family system or made external attributions in that regard. Future research could help to shed light on individualised vs. family embedded interpretations of life trajectories.

Regarding gendered experiences of retirement, men generally experience retirement more positively than women. We then discuss possible factors that may help to understand these results. For example, just as men and women hold different work and life trajectories, in general, they may also have different ways of adapting and experiencing retirement [6,24], and women may have more difficulty in adapting to this phase of life [56]. Additionally, there are studies reporting that men experience higher levels of satisfaction in retirement, while women begin retirement with more depressive symptoms and more unfavourable attitudes towards ageing itself [6,45]. In line with these findings, women reported a predominance of negative experiences in this phase, marked by sadness, frustration, and lack of life motivation. These feelings emerged in their responses, and they were closely associated with ageism and the respective negative view of age. As Rosa stated [62], there is a negative view that associates old age with death, referring to it as the last phase of human life. In this view, physical and cognitive losses overwhelm future projects and foster the feeling of loss in relation to the past, experiencing feelings of unhappiness, frustration, and lack of motivation [63,64]. We also found that the ageist attitudes commonly accepted in our society largely influence how women experience retirement, negatively affecting their self-image and behaviour [65,66]. For some of the women interviewed, retirement turned out to be an event that represents the entrance into old age and, consequently, all that this implies in our society, namely, illness, declining abilities, and death. A relevant angle is to analyse these statements in line with an internalisation of both age-based and gender-based prejudice. Knowledge about intersecting disadvantaged social categories is scarce [67], and our results seems to suggest an interesting interplay between age and gender. For women, this phase of life is the target of a double source of prejudice: on the one hand, ageism equates old age with illness, death, and decline in abilities, influencing the expectations that older women have about the ageing process itself [65,66]; on the other hand, sexism leads to a gendered distribution of occupations, where women undertake, once again, domestic and care work [1]. An example of this age/gender intersection is the time spent with grandchildren. On the one hand, it can contribute to the feeling of being “imprisoned” for some female caregivers after retirement (thus implying a double burden of age and gender); on the other hand, it can be a source of positive experiences, bringing purpose to the lives of those who, for example, have health problems (thus, gender expectations represent a buffer for ageism). Our study opens avenues for understanding the complex role of intersectionality with other social categories when approaching a phenomenon from a gender perspective.

Men stated that retirement is meeting their expectations, showing positive experiences characterised by freedom, new occupations, new interests, and hobbies. In general, men felt free, relieved, uncommitted, and satisfied. Women’s dissatisfaction with retirement can also be understood by the fact that their occupations do not match their expectations (e.g., traveling instead of housework). The prominence of domestic and caregiving activities can lead women to feel overburdened and “homebound” [4], compared to men, who feel freer and are more involved in activities of personal interest, such as hobbies. This leads to the fact that retirement can be much more attractive for men, who are not forced to perpetuate at this stage of life most of the concerns they had before retirement [58]. Additionally, the fact that the women interviewed have less time post retirement than men, may lead to the fact that they are still in a phase of disenchantment/reorientation compared to men, who may already be in a phase of stability, where a routine is established and tasks are performed that allow a sense of usefulness to oneself [54].

Apart from the intersection between disadvantaged situations, the results allowed us to propose that not only actual diseases, but also crises and situations that limit people’s lives in general can be detrimental to achieving the anticipated fruits of retirement. The COVID-19 pandemic represented such an obstacle to active retirement: it came as an added challenge for all those already retired and impacted the lives of older people in many ways [1]. Many people saw their economic stability, mental and physical health, as well as their social networks threatened [26]. For example, the onset of illness limited and/or diminished the routine of several respondents, and even appeared as an impediment to the plans and prospects they had for retirement. Also, retired people were largely affected by the pandemic. If people went out regularly to do various social activities outside before, with the pandemic, they were no longer able to do so, as from its inception the elderly population was urged to stay at home as much as possible until the vaccines emerged and were administered [68]. Most interviewees reported a decrease in their outdoor routines, which occurred, essentially, through the reduction in the time allocated to socialization activities and hobbies [69,70], those activities being a source of pleasure, life satisfaction, and well-being.

In relation to the last objective of the study, we could see that, in general, the men and women interviewed experience retirement quite differently. We found that they not only have different occupations, but also different perceptions of pensions and general experiences of this stage of life. Overall, retirement proved to be more positive for men than for women. Only the COVID-19 pandemic and the onset of illness were shown to hinder active retirement among men and women alike. In sum, these results show that, like the study of ageing, retirement should be understood in light of a life course perspective, analysing the role of the different employment and life trajectories of men and women, as well as the context in which they take place.

## 5. Conclusions

This research has provided important contributions to the study of gender and retirement, increasing our knowledge about how men and women live their retirement. It not only increased the number of Portuguese studies on this theme, but we believe to have provided some knowledge that can be deepened by other researchers. We found that retirement is not immune to gender issues, highlighting how the different life and work trajectories of men and women influence this part of their life. Still, we hope that more studies will continue to explore these issues, as gender equality is an ongoing achievement. It is urgent to increase knowledge in this area, so that retirement is not seen as the “beginning of the end”, but rather as another opportunity for personal and social life development, similar to those that arise throughout the life course.

This study has some limitations that should be considered. The retrospective nature of the study is one of these, since the perspectives on retirement addressed by the people interviewed may be the result of a reconstruction of reality, which we tend to do in order to lessen the impact when something does not go as we plan. Another limitation is that this is a qualitative study with a small sample that is limited to one geographic area, not allowing the results to be generalised to society in general. Although the occupations before retirement can be comparable to studies conducted in more urban areas, e.g., [46], the available activities in rural areas might be different (e.g., fewer possibilities for gardening in urban areas).

More studies are needed to understand how retired people experience this phase of their life. We even suggest that a longitudinal study be conducted, with a larger sample size and with retired people from different areas other than the rural one. For example, this could be research that analyses the different employment trajectories of men and women, while tracking the expectations and perspectives of people in the pre-retirement phase until the time of transition and adaptation to retirement. With such a line of research, it will be possible to analyse those trajectories and their impact on retirement, as well as people’s experiences and their relation to prior perspectives regarding that important moment in life.

## Figures and Tables

**Table 1 behavsci-13-00774-t001:** Number of the person interviewed, sex, characterization of the type of retirement, reason why they retired, and how they use their time.

Interview Number	Sex	Type of Retirement	Retirement Time (Years)	Reason for Retirement	How Time Is Spent
1	Man	On legal time	12	I’ve reached the required work time length	Walks
2	Woman	Early	13	Tiredness	Exercise, reading, taking care of the grandchildren
3	Man	Early	12	Exhausting work	Walks, gardening, agriculture, conviviality
4	Woman	On legal time	6	I’ve reached the required work time length	Help their husband
5	Woman	Late	6	I’ve reached the required work time length	Gardening, reading
6	Man	On legal time	28	I’ve reached the required work time length Tiredness	Walking, being with family
7	Man	On legal time	7	Tired of the factory environment	Agriculture, gardening
8	Woman	On legal time	16	The opportunity arose; I’ve reached the required work time length	Housework
9	Woman	Early	2	Health problem	Being with their grandchildren, housework, aqua aerobics
10	Man	Early	12	Health problem	Leisure, travelling, Reading
11	Man	On legal time	8	I’ve reached the required work time length	Occasional work for the City Council
12	Woman	On legal time	10	I’ve reached the required work time length	Housework, aqua aerobics, walks, outings, caring for grandchildren
13	Woman	Early	16	The opportunity arose	Housework, gardening, reading, watching films
14	Woman	On legal time	8	Employment subsidy time has ended	Gardening, walking, computer
15	Man	On legal time	20	I’ve reached the required work time length	Agriculture, gardening
16	Man	On legal time	22	I’ve reached the required work time length	Gardening, home agriculture

**Table 2 behavsci-13-00774-t002:** Map of themes and sub-themes.

Themes	Sub-Themes
From the absence of gender perspectives to men’s perspectives on retirement	Lack of perspectives on retirement
	Shared perspectives—freedom and travel
	Men’s perspectives on retirement
From gendered occupations in retirement to shared occupations	Feminine occupations
	Masculine occupations
	Shared occupations—gardening
Gender differences in perceptions of retirement pensions	Women consider their pensions as being low
	Men are more satisfied with their pensions
Gendered experiences in retirement	Women’s experiences
	Men’s experiences
Obstacles to an active retirement	Decline in physical and psychological health
	Negative impact of the COVID-19 pandemic

## Data Availability

The data presented in this study are available on request from the corresponding author. The data are not publicly available due to ethical restrictions.

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
