# Peer review of "The Gender of Retirement in a Double-Ageing Country: Perspectives and Experiences of Retired Women and Men in Portugal"

_behavsci, 2023, doi:10.3390/bs13090774_

Round 1
Reviewer 1 Report
I had the opportunity to review the manuscript titled "The Gender of Retirement: Perspectives and Experiences of Retired Women and Men." I commend the authors for addressing an important topic, but I have several suggestions that could enhance the clarity and impact of the paper.
1-Introduction Length and Content: The introduction is considerably lengthy and contains a substantial amount of known information. I recommend streamlining this section by focusing on context-specific information about Portugal. By allocating more space to relevant context and the unique aspects of retirement experiences in Portugal, the introduction can more effectively engage the readers and set the stage for the study.
2-Title and Context: The title of the study could benefit from reflecting the specific context of Portugal, thus encapsulating the essence of the research more accurately. This adjustment would provide readers with a clear understanding of the study's scope and relevance.
3-Section Structure and Synthesis: Sections 1.3 ("Life Course Perspective") and 1.4 ("Gender") as well as sections 1.5 ("Gender: The Different Trajectories of Men and Women") could be synthesized more effectively. Merging these sections could help in presenting a cohesive narrative that highlights the gender-related trajectories within a broader life course perspective. This would provide a more integrated understanding of the subject matter.
4-Recent References and Research Gap Justification: The references cited are relatively dated, which could potentially weaken the justification for the research gap and the significance of the study. I suggest the authors revise the literature review to include more recent studies that support the relevance of investigating retirement experiences and gender dynamics in Portugal. This would also help contextualize the study within the current scholarly discourse.
5-Expanded Significance of the Study: The last paragraph of the introduction outlines the reasons why the study is needed in Portugal. To enhance the impact of this part, the authors should consider expanding it and providing a stronger foundation through recent literature. Clear references to current social, economic, and gender-related trends in Portugal would underscore the importance of the research.
6- The manuscript lacks clarity regarding who conducted the interviews and how the data collection was managed. This information is pivotal for readers to understand the rigor of the study and the potential impact of the interviewers' characteristics on the responses obtained. Providing details about the interviewers' qualifications, training, and their roles in the research process would enhance the transparency of the study. Furthermore, a comprehensive description of the recruitment process would elucidate how participants were selected and approached for the study. Information about the criteria for inclusion, the sampling strategy, and the number of participants interviewed would strengthen the methodology section and provide a clear picture of the study's scope.
7- The themes presented in the paper appear to reflect well-known and somewhat clichéd information. I encourage you to reconsider these themes and strive for more novel and nuanced insights. It's crucial to ensure that the themes explored in the study contribute substantially to the existing body of knowledge. While I understand that the introduction touched upon some of these themes, the main body of the paper should delve into fresh perspectives that shed light on new aspects of the subject matter. I recommend revisiting the data with a critical eye to identify untapped nuances and less explored angles. This approach could lead to the discovery of themes that provide a more original and insightful contribution to the discourse surrounding gender and retirement. By doing so, your study will not only stand out but also enrich the field with valuable, distinct viewpoints.
8- In Table 1 of your manuscript, it would greatly benefit the readers if you could include a column indicating the "Time Since Retirement" for the participants included in the study. This addition would provide a crucial dimension to the information presented in the table and allow readers to understand the timing of the retirement experiences being discussed.
9- It's evident that several of the references utilized in the Discussion are significantly outdated, some exceeding 20 and 25 years. Relying on such aged references could potentially weaken the impact and credibility of your study. To maintain the relevance and significance of your findings, it's crucial to anchor your discussion in the current literature.
Thank you for considering my feedback,
Author Response
In the attached file you will find the answers to your suggestions.
Kind regards

Reviewer 2 Report
Manuscript 2553991: "The gender of retirement: Perspectives and experiences of retired women and men"
The current papers analysis retirement experiences of women and men in Portugal. Using qualitative methodology, the authors find gender heterogeneity, and different trajectories by gender.
In general the paper is relevant, but the introduction does not explain well enough the motivation and importance. Also, conclusions are somehow deviating from the introduction.
I have some doubts about the sample selected, as discussed below. The gender aspect could be explained better.
I wonder whether some more quantitative data analysis could complement the qualitative results presented.
1. The first paragraph starts discussing aging population in Europe, but does not provide any figures for Portugal. It would be more appropriate to provide details of aging population of Portugal in Europe. Is it the same, more or less?
2. I think the paper could have been motivated better: the second paragraph of the introduction claims: ”Given the increase in the population at or near retirement age, it is increasingly relevant to understand when and why people retire, what they do, and what they actually experience after retiring (Beehr & Bowling, 2012).” Why exactly it is relevant has not been explained. Is because of costs, is it because of social isolation or what? This is relevant and should be explained well.
3. Introduction, the sentence: “..this topic will also allow demystifying existing exogenous beliefs in relation to this stage of life, which is still neglected and considered one of the least interesting stages of human life (Fonseca, 2011).” I do not understand this sentence, especially exogenous beliefs.
4. Part 1.1. First sentence, I am not sure why it needs to be said that there is a multiplicity of definitions for the concept of retirement. The one provided is probably the most official so just say.
5. Last sentence of 1.1 refers to various reasons for retirement but not to the age and years of legal contributions for pensions. This should have been mentioned.
6. The gender aspect is indeed very relevant, but again it is not explained well enough why we should expect differences between men and women.
7. Age of people interviewed seems quite wide, from 59 to 88. I expect those close to 88 retired long before that age. So I am not clear about this choice.
8. Similarly, I am not sure why you would want to interview pensioners that retired 20 years ago. I would have made more sense to interview people who had just retired, for example. Since the sample looks very heterogeneous.
9. The abstract and introduction does not really refer to the impact of Covid19 on retirement, but the conclusions clearly do. That creates a bit of confusion over what exactly if the focus of the paper and its contribution. Are you considering retirement over pandemic? IF that is the case, if should be made clear up front.
10. The authors rely too much on findings from existing studies. In fact 54 studies cited are a bit too much.
English needs some editing.
Author Response
Answer to
Comments and Suggestions
from Reviewer 2
Kind Regards

Round 2
Reviewer 1 Report
The authors' response to the feedback provided was satisfactory, resulting in significant improvements to the manuscript. The revisions made, including streamlining the introduction, refining the title, synthesizing sections for better coherence, updating references, enhancing the methodology section, and providing additional context in the discussion, have collectively strengthened the paper. These changes not only address the initial concerns but also enhance the manuscript's clarity, relevance, and overall quality, making it a more valuable contribution to the field.
Reviewer 2 Report
Dear Authors
thank you very much for all your additional effort. The paper has very much improved and has a much better shape.
My concerns have been addressed.